# Selectively Biodegradable Polyesters: Nature-Inspired Construction Materials for Future Biomedical Applications

**DOI:** 10.3390/polym11061061

**Published:** 2019-06-19

**Authors:** Tomáš Urbánek, Eliézer Jäger, Alessandro Jäger, Martin Hrubý

**Affiliations:** Institute of Macromolecular Chemistry, Czech Academy of Sciences, Heyrovského náměstí 2, 162 00 Prague 6, Czechia; urbanekto@gmail.com (T.U.); jager@imc.cas.cz (E.J.); ajager@imc.cas.cz (A.J.)

**Keywords:** polyester, polycondensation, ring-opening, stimuli-sensitive, drug delivery, biodegradability, medical application

## Abstract

In the last half-century, the development of biodegradable polyesters for biomedical applications has advanced significantly. Biodegradable polyester materials containing external stimuli-sensitive linkages are favored in the development of therapeutic devices for pharmacological applications such as delivery vehicles for controlled/sustained drug release. These selectively biodegradable polyesters degrade after particular external stimulus (e.g., pH or redox potential change or the presence of certain enzymes). This review outlines the current development of biodegradable synthetic polyesters materials able to undergo hydrolytic or enzymatic degradation for various biomedical applications, including tissue engineering, temporary implants, wound healing and drug delivery.

## 1. Introduction

The days of Carothers’s skepticism about the quality of polyesters (PES) are long gone and deeper knowledge brings insight into many areas of interest every day. Due to the extensive research, the polymeric structure and the final material properties can be deliberately fine-tuned by different approaches of polymer chemistry for the purposes of specific applications [1].

The polymer chemistry has a wide toolbox available for tuning and fine-tuning the structure and properties of final material. The nature of monomers, the regulation of the molar mass and the dispersity are the very basic concept of the control over properties. Nowadays, there are several polymerization techniques, which enable us to control molar mass and narrow dispersity very precisely. The introduction of branching points into the polymer structure is another simple way of modulation of the final properties [2]. There are many other approaches of modulating the final properties of polymer materials—introduction of functional groups, combination of two or more polymers or with other materials, etc. The keystone of successful fine-tuning is the understanding of the relationship between the structure and functional properties of polymer materials.

In addition, there are particular polymers—the so-called smart polymers—which benefit from the fact that they can undergo structural changes after being exposed to an external stimulus (pH change, temperature change, presence of particular molecules, etc.) leading to significant change in physico-chemical or solution properties within a relatively narrow window of the external environment change. Obviously, there are such smart polyesters, which respond to external stimuli, too.

### 1.1. Synthesis and Mechanisms

Albertsson and Varma [3] reported that polycondensation, ring opening polymerization (ROP) and enzymatic polymerization are the three major routes for the synthesis of aliphatic polyesters. A very interesting but minor polymerization technique that also leads to polyesters is the free radical ROP of 2-methylidene-1,3-dioxepane. This process offers us a wide range of copolymers combining ester bond with repeating units derived from vinylic monomers in the backbone, which would not be possible to prepare by any other process [4,5]. The synthetic routes leading to aliphatic polyesters are schematically shown in Figure 1.

Polycondensation is characterized by a stepwise polymerization reaction involving difunctional monomers of the AB type, i.e., hydroxy carboxylic acids, or from a combination of AA and BB difunctional monomers resulting in the formation of a small byproduct, e.g., water [6,7]. Basically, examples of polycondensation reactions, which proceed using difunctional monomers are the esterification of diacids with diols, diacid chlorides with diols or the ester interchange reaction of diols with diesters (Figure 2) [3,8,9,10].

The pioneering studies of Carothers in the 1930s [11,12] on kinetics of polycondensation provided the first insights on the fundamentals analysis of step polymerization kinetics. It was defined by his equation that high molecular weight polymers (${\bar{X}}_{n}$ > 50) could be achieved only at a very high degree of conversions (*p* > 98%–99%).
(1)$${\bar{X}}_{n}=\frac{1}{1-p}$$

Therefore, the formation of the ester group during the polymerization process is characterized by an equilibrium reaction (Figure 3) and in order to prepare high molecular weight polymers, two major prerequisites must be fulfilled [13,14]. Firstly, the equilibrium constant of polycondensation (*K_P_*) must be sufficiently high and secondly, according to Equation (2) in case of heteropolycondensation (e.g., dialcohols and dicarboxylic acids) the stoichiometry (1:1) must be strictly obeyed.

The number average degree of polymerization (${\bar{X}}_{n}$) in this case is associated to *K_P_* through the derivation of Equation (2) and *K_P_* values around 10 were generally found for a condensation reaction of aliphatic alcohols with carboxylic acids [7].
(2)$$K_{p}=\frac{\left[-C\left(O\right)O-\right]\left[H_{2}O\right]}{\left[-C\left(O\right)OH\right]\left[HO-\right]}$$

Thus, according to Equation (3) values of ${\bar{X}}_{n}$ around 4 for *K_P_* ≈ 10 drives the polymerization reaction to the equilibrium.
(3)$${\bar{X}}_{n}=K_{p}^{0.5}+1$$

In general *K_P_* must be increased to values higher as 2400 during the reaction since polyesters with ${\bar{X}}_{n}$ > 50 is required to fulfill the basic physical properties of the polymer. In this case removal of the by-product (e.g., water) [15] from the reaction, the use of a catalyst [16,17,18] and high temperature settings (180–250 °C) [19,20] are some of the usual strategies applied in order to drive the reaction equilibrium toward high *K_P_* values resulting in higher conversion rates. However, one of the main drawbacks in the polycondensation reaction is given by Equation (4). According to it the increase in the conversion (p) leads also to the increase in the polymer dispersity (${\bar{X}}_{w}/{\bar{X}}_{n}$). Thus, at high conversions (*p* > 98%–99%) the polymer ${\bar{X}}_{w}/{\bar{X}}_{n}$ has the tendency to approach 2 as a value [11,21].
(4)$$\begin{array}{c} {\bar{X}}_{n}=\frac{1}{1-p}\\ {\bar{X}}_{w}=\frac{1+p}{1-p} \end{array}\;\}\;\frac{{\bar{X}}_{n}}{{\bar{X}}_{w}}=1+p$$

Nowadays, is with no doubt that ROP polymerization (e.g., of cyclic glycolide, lactides and lactones, Figure 4) is one of the most employed polymerization routes in order to circumvent the drawbacks from the polycondensation. The ROP polymerization processes several advantages compared to traditional condensation polymerization, e.g., high conversion without the necessity of removal of reaction byproducts, shorter reaction times, mild synthetic conditions and the use of a stoichiometric balance of monomers. Moreover, the ROP polymerization sometimes proceeds in a “living” manner, without side reactions, which allows good control of the polymer characteristics (narrow molecular weight distribution and predictable final molecular weight of polymer) [6,10,22]. All these remarkable advantages make ROP the method of choice for the preparation of high-molecular weight aliphatic homo- and copolyesters.

ROP is a very flexible synthetic route and allows the use of several mechanistic approaches such as cationic, anionic, as well as coordinative catalyst or initiators have been reported [23,24]. In general, ionic (free ions and non-bulky ion pairs) are much more reactive and in the case of polyester inter and intra-molecular transesterification occurs lowering the molecular weight and broadening the molecular weight distribution of the polymer [10,25]. Otherwise, organometallic compounds of metals (Al, Sn, Ti or Mg) are more energetically favorable in comparison to their anionic counterparts and able to provide full control to the polymerization. Taking into account the several mechanisms involved in the ROP the two major proceeds using organometallics that are acting as initiators or as catalysts. In the cases where it is used as an initiator the polymerization proceeds through an “insertion-coordination” mechanism (Figure 5a) and when it is used as catalyst (Figure 5b) the polymerization is initiated by any nucleophile present in the polymerization medium.

The third route to obtain polyester under mild conditions is using enzymatic polymerization. This method avoids the use of toxic reagents and allows the recycling of the catalyst. Moreover, regional and stereo selectivity of enzymes allows the direct synthesis of functional polyesters avoiding the use of protected monomers and block copolymers. However, the production of polymers with relatively low molecular weight is still a major drawback in the application of enzymatic polymerization on the synthesis of polyesters.

Nowadays, aliphatic polyesters constitute one of the most important classes of synthetic biodegradable and biocompatible polymer intended for biomedical applications [10,26,27]. Most of them are commercially available in several types, some examples of FDA-approved aliphatic polyesters are polycaprolactone (PCL) [28,29], poly(glycolic acid) (PGA), poly(l-lactic acid) (PLA) [30,31] and poly(lactic-co-glycolic acid) (PLGA) [32]. They have been extensively studied for their biocompatibility [33,34,35,36], biodegradability [37,38] and bioresorbability [39]. It is very well established that they are highly biocompatible materials [40], easily hydrolysable into human body [41,42,43] and therefore they can be used for biomedical applications in the production of drug carrier devices for controlled release [44,45,46,47]. Between the FDA-approved synthetic biodegradable and biocompatible polymers, poly(alkylene succinate) based polyesters has also proven to be an interesting alternative on the production of biomaterials for a myriad of applications [15,48,49,50,51,52,53]. Since they are free of cytotoxic degradation products, e.g., succinic acid, which is an intermediate in the TCA cycle (tricarboxylic acid cycle, citric acid cycle), makes poly(alkylene succinates) potential candidates as biomaterials on the development of drug delivery structures [54,55,56,57]. Moreover, the copolymerization of alkylene succinates with fatty acids (naturally occurring body compounds) [58,59], such as dilinoleic acid, allow the preparation of more hydrophobic biodegradable polymers. These more hydrophobic polymers are suitable for fine-tune hydrophobic drugs encapsulation via polymer–drug interactions when, for example, these polymers are used as drug nanocarriers [8,9,53,60,61].

### 1.2. Drug Release from Polyesters in General

The potential to protect the active drug component from degradation together with sustaining their release, as well as the capability to modulate the active drug diffusion and polymer degradation resulting in inert body-friendly degradation byproducts, make polyesters successful candidates for drug delivery systems in biomedical applications. The encapsulation of drugs into polyester-based nanosystems, subsequent drug release and to some extent also polymer degradation are generally dependent on the polymer/matrix interaction with the dissolved (compound at the amorphous state) or dispersed (compound at crystalline state) drug [8,62,63]. This is directly related to the solubility (for the crystalline host molecules) and/or miscibility (for the amorphous load) of the drug with the polymer matrix and vice-versa. The polymer crystallinity can also play an important role for the encapsulation, drug release and degradation. Some studies reported that the drug loading decreased with the increase of polymer crystallinity and the glass transition temperature (*T*_g_) [14,63,64,65], other studies reported that the drug loading capacity for polyester nanoparticles can be significantly enhanced by hydrophobic effects between the polymer and the drug and in combination with hydrogen bonding, electrostatic interaction and dipole–dipole interactions modifying the drug release profile [66,67,68]. In addition, the degradation behavior of the polyester matrix is an important prerequisite for the potential drug release. There are two main mechanisms involved in the degradation of polyesters and they are dependent on the relative rates of water diffusion into the polymer matrix and on the polymer degradation rate [8,62]. In the case where the rate of polymer degradation is faster than the rate of water diffusion into the polymer matrix the mechanism is called surface degradation. On the contrary, when diffusion of water into the matrix is faster than polymer degradation and the whole matrix is affected by degradation and erosion, the process is called bulk degradation. In general, under biological conditions (in vitro and in vivo) the degradation of polyesters without specific responsive linkage (described in detail hereafter) proceeds by random hydrolytic cleavage of ester linkages.

## 2. Polyesters with Stimuli-Sensitive Linkages

### 2.1. pH-Labile Polyesters

The pH-labile polyesters have found a wide range of applications in drug release for diseases where certain pH shift from the physiological pH is observed. Typically, the polymer degradation may be selective in this way in cancer and inflamed tissues (in the case of inflammation, the presence of reactive oxygen species is also used for specific drug release or polymer degradation, see below) [62]. While in the healthy tissues the pH value is around 7.4, in the cancer and inflamed tissue the pH value is significantly lower—around 6.5 [69]. It has also been shown that some cell compartments such as lysosomes or endosomes exhibit lower the pH to between 5 and 6 [70,71]. Many pH-labile systems have been developed so far and there is always a pH-sensitive functional group present in the structure. The most common bonds with such properties are the hydrazone bond [72,73,74,75,76,77], amine/imine groups [78,79], acetals/ketals [80,81], ortho esters [82,83], *cis*-aconityl group [84,85], etc. The selected pH-sensitive linkages are shown in Figure 6.

The well-known pH-sensitive linkage hydrazide/hydrazone has become one of the most popular pH-sensitive linkages in drug delivery systems. The pH-sensitivity in compact biodegradable aliphatic polyester dendrimer has been observed by near-infrared (NIR) fluorescence [77]. The NIR fluorophore cypates have been attached to the PEGylated polyester dendrimer via a hydrazone bond. The hydrazone bond is stable in neutral pH so the dye sits on the dendrimer. The activation of the NIR fluorescence occurs when the dye is cleaved off the dendrimer in the acidic pH. In addition, the dendrimer scaffold can be enzymatically degraded by endogenous esterase.

The PES-like polyketal particles have been described for the controlled drug release systems. Due to their easy synthesis they hold promises for further development. The system based on poly(1,4-phenyleneacetone dimethylene ketal) has been designed as a pH-sensitive drug carrier. It has been shown that the degradation time of the polyketal backbone undergoes three times faster in acidic conditions (pH 5.5) compared to the physiological conditions (pH 7.4). In addition, the loading with the hydrophobic drug has been successfully done, so the loaded biodegradable drug vehicles have been proposed for therapeutic treatment involving phagocytosis by macrophages [86]. 

The pH-lability of orthoester bond has been described in the literature [87]. The current trend of designing very specific systems for individual applications has motivated the research group at the University of Minnesota to prepare double responsive copolymer where along the pH sensitivity due to the orthoester group, the amide group is incorporated to introduce the temperature-responsivity, too. The sol–gel transition temperature (Tt) is highly depending on the polymer structure so the Tt can be set up precisely and the pH-lability in the acidic environment is maintained [83].

The comparison of pH lability of hydrazone and *cis*-aconityl linkage has been done in the study where the doxorubicin was attached to the diblock copolymer composed of poly(l-lactic acid) and methoxy-poly(ethylene oxide) via these pH labile bonds and the degree of release has been evaluated [85]. Although both linkages have cleaved in the acidic environment, the *cis*-aconityl bond showed higher lability, which led to faster drug release. However when the drug was measured at pH 7, the *cis*-aconityl bond exhibited higher stability.

### 2.2. Reductively Labile Polyesters

There are two main approaches for the construction of reductively degradable drug delivery systems. They utilize (i) the reduction of the disulfide bond by glutathione specifically in cancer cells and (ii) the bacterial reduction of aromatic azo-dye, which allows for colon-selective degradation [88,89].

The azo group is well known for the photosensitive properties in organic chemistry. The *Z*-configuration can be easily switched to a more stable *E*-configuration by specific wavelength irradiation [90]. In the polymer chemistry, the reduction of azo group is taken in advantage and it is widely used for drug delivery systems. It highly depends on whether the azo linkage is part of the backbone or it occurs in the side group for the reduction. It has been observed that the azo linkage incorporated in polymeric backbone degrades slower [88,89,91]. The difference in the degradation mechanism has been noticed when aromatic (Figure 7) and aliphatic polymers were compared—the aliphatic azo compounds such as azobis(isobutyronitrile) undergo a rather nitrogen-releasing thermal or photoinduced decomposition than bioreduction. The azo bond is known for the ability to be reduced by intestinal microflora [92]. There are also polyester systems containing a reductively labile azo group. These polyesters containing the azo linkages can be used in colon-specific therapy [93,94,95]. Most precisely, the aromatic poly(ether-ester) has been synthetized by classical polycondensation of 4-[(4-hydroxyphenyl)azo]benzoic acid and its derivatives. These poly(ether-ester)s are well degraded by azoreductase—enzymes located in the intestinal bacteria and therefore they were suggested as the carrier for colon-specific drug release [93].

A considerable number of systems have been invented for drug specific release at the cancer tissue with the use of the disulfide bond. The reason why the disulfide bond is often used is that the cancer environment shows a more reductive potential compared to healthy tissue due to lower oxygen partial pressure in such tissues. This shift in redox potential is connected with higher concentration of reduced glutathione (GSH) [96]. In general, also the intracellular environments exhibit higher concentrations of GSH compared to the extracellular (1–10 mM vs. 1–10 μM) [97]. While GHS occurs in the reduced form in the intracellular compartments, in extracellular rather in the oxidized form [98]. Therefore this system based on GSH has been widely used not only in target drug release [99,100] but also in gene delivery [101,102,103,104].

There is more than one physiological function of GSH in organisms. Beside it being an antioxidant agent, which prevents the damage caused by reactive oxygen species, it also serves as the reducing agent, particularly for the disulfide bond [105]. The GSH donates its electrons to the proteins connected by a disulfide bond or to the protein, which has an intramolecular disulfide connection and oxidizes itself to glutathione disulfide. This phenomenon can be taken advantage of in drug design and many researchers have suggested polyester systems containing a disulfide bond as a drug carrier aiming at specific release or polymer degradation on cancer cells. The mechanism of the GSH oxidation in the presence of a polyester drug carrier containing disulfide linkage is shown in Figure 8.

The strategy of Shen’s research group was to prepare such micelles, which would be stable in the blood stream (would not disintegrate into unimers there) so the disintegration will occur in the cancer cells, specifically. They design stimuli sensitive crosslinkable micelles based on a hydrophobic polyester core with reversible disulfide crosslinking points and hydrophilic poly(ethylene oxide) shell. Their synthetized micelles loaded with paclitaxel showed high stability in circulation and due to the cytoplasmic glutathione, the paclitaxel release was targeted in cancer cells, preferably [106].

In the study of Prof. Farokhzad et al. [107], the system based on poly(ether ester) with introduced disulfide linkages was compared with a system where the disulfide linkages were missing. Even though the systems exhibit very similar properties such as the ability of nanoparticle formation and dye and drug encapsulation, the system containing disulfide linkages was much more sensitive to the introduced reducing agents and thus the dye release was much faster. Not surprisingly, the cell viability was reduced by the drug-loaded reduction-sensitive nanoparticles compared to reduction-insensitive nanoparticles with a comparable amount of drug [107].

### 2.3. Reactive Oxygen Species (ROS)-Labile Polyesters

The importance of ROS that it is involved in several pathological sates and cellular signalization have attracted great attention towards the development of chemical tools for specific delivery to ROS-rich niches, and ROS-responsive micro-or nano-delivery systems. Delivery systems that are capable of targeting and releasing active molecules (chemotherapeutics, antigens, adjuvants, antioxidants, proteins, etc.) at sites of high ROS expression have the potential for high biological impact. The capability to generate a triggered polymer response (e.g., to deliver cargo or to induce polymer self-immolation/degradation) at ROS rich sites has received particular interest, e.g., for the preparation of delivery system targeting the tumor microenvironment and inflammation sites. Several ROS-responsive polyesters were designed based on oxidatively cleavable linkers such as thioethers, selenium-containing bonds, phenylboronic acid esters and aryl oxalates—schematically shown in Table 1. In this section, we will systematically describe ROS-responsive polyesters that have been mostly developed so far from those moieties and their nanoparticles (NPs) preparation and characterization with the proof-of principle tested in in vitro and in vivo models.

#### 2.3.1. Poly(propylene sulfide)s

Poly(propylene sulfide)s have been one of the most investigated classes of ROS-responsive polyesters. Due to their very low dipole moment, in the presence of oxidative conditions, poly(propylene sulfide)s containing polymers undergo a phase transition from hydrophobic sulfide to a more hydrophilic sulfoxide or sulfone [108,109]. Oxidation, for example, by H_2_O_2_, would be a trigger that converts the polyester from hydrophobic to hydrophilic resulting in morphological transitions, swelling, solubilization or release of the entrapped molecules.

Poly(propylene sulfide)s can be produced by several techniques such as propagation reactions producing main-chain thioethers from thiols or from cyclic strained thioethers comprising step-growth polymerization mechanism or most recently by “living’ radical polymerization such as the ROP of episulfides that provides amphiphilic block copolymers [108]. One example is the pioneer synthesis of symmetric amphiphilic triblock copolymer of ethylene glycol and propylene sulphide (named PEO-PPS-PEO) described by Hubbell’s group in 2004 [110]. The triblock copolymer comprises the hydrophobic block of poly(propylene sulphide) (PPS), owing to its extreme hydrophobicity, its low glass-transition temperature and most importantly its oxidative conversion from a hydrophobe to a hydrophile, poly(propylene sulphoxide) and ultimately poly(propylene sulphone) undergoes self-assembly into polymer vesicles where the presence of H_2_O_2_ induces a morphological change of otherwise highly stable vesicles to worm-like micelles, then to spherical micelles and ultimately to non-associating unimolecular micelles. Poly(propylene sulfide) nanoparticles also can be directed synthesized by anionic ROP emulsion polymerization [111,112]. By this synthetic strategy Allen et al., synthesized poly(propylene sulfide) NPs with the addition of hydrophobic solvatochromic dyes such as Reichardt’s dye or fluorescent dyes such as Nile red dye during the polymerization process resulting in their stabilization in aqueous media by encapsulation. Subsequent addition of H_2_O_2_ or low levels of sodium hypochlorite (NaOCl), causes particle oxidation followed by particles swelling and release of cargo. Furthermore, it was demonstrated that this oxidative release could be facilitated through peroxidase enzyme oxidant generation. Chloroperoxidase (CPO) and human myeloperoxidase (hMPO) enzymes in the presence of 200 mM NaCl and 500 μM H_2_O_2_, generate hypochlorous acid (HOCl), which oxidizes the poly(propylene sulphide) NPs and causes the release of the encapsulated cargo. Taking advantage of this oxidation-mediated transition, various ROS-sensitive structures based on polysulfides such as micelles [113,114], microspheres [115], vesicles [110] and hydrogels [116] have been synthesized and their proof-of-principle tested as delivery agents for antioxidant [115], chemotherapeutic [117] and immunomodulatory [113,118] compounds.

#### 2.3.2. Selenium-Containing Polyesters

Diselenide-containing polymers have been developed because they attractive rapid ROS response kinetics and their sensitivity to oxidation–reduction, which makes them promising candidates for a dual redox response. Similar to polysulfides in thioether-containing polymers (aforementioned), hydrophobic selenides are able to be oxidized to water soluble selenoxides and selenones [119]. In addition, selenium-containing polymers possess a higher sensitivity to oxidants because of the weaker bond energy of the C–Se bond compared with the C–S bond.

Usually selenides are prepared by step-growth polymerization synthesis resulting in polyester urethanes (polyurethanes) as block copolymers. One pioneer study was designed by Ma et al., where a diselenide containing polyurethane block was synthesized via polymerization of toluene diisocyanate in excess with diselenide-containing diols and posterior termination by PEG monomethyl ether. The new amphiphilic diselenide-containing triblock copolymer of ABA-type (named PEG-PUSeSe-PEG) could self-assemble in micelles cargo doxorubicin or rhodamine B with unique dual redox responsive behavior [120]. The loaded cargo was released either in the presence of H_2_O_2_ (0.01%) or glutathione (0.01 mg/mL) within a period of 5 h. In forthcoming work, these diselenide-containing triblock copolymers (PEG-PUSeSe-PEG) were also shown to be responsive to singlet oxygen [121]. Upon irradiation of PEG-PUSeSe-PEG micelles in the presence of porphyrin derivatives with a red light, the generated singlet oxygen oxidized the diselenide-containing block copolymers and cleaved the diselenide bonds resulting in the disassembly of the micelles and the release of the loaded Dox.

Selenide based polyesters were also prepared by ROP. Recently Yu at al., demonstrated the ROP of caprolactone for the preparation of selenide ROS-responsive poly(ε-caprolactone) (PCL)-type polyesters with pendant selenide motifs. The NPs prepared from this newly synthesized polyester containing pendant selenide groups respond much faster to H_2_O_2_ than NPs prepared by the same pathway (ROP of caprolactone), however containing thioether as pendant groups, most likely due to the superior sensitivity of the selenide pendant groups towards H_2_O_2_ [122]. In a straightforward approach, the synthesis of a variety of multi-responsive linear and cross-linked diselenide-containing polyesters was designed by Wang et al., from a novel one-pot two-step process consisting of a nucleophilic ring opening reaction of *γ*-butyroselenolactone with a wide range of alcohols, followed by a stepwise polymerization in situ ring opening/oxidation process [123]. By this pathway diselenide bonds can be easily introduced into several types of polyester architectures, avoiding thorough synthesis procedures and including dynamic diselenide intermediates.

#### 2.3.3. Aryl Boronic Esters Containing Polyesters 

Among the aforementioned oxidation responsive polymers, the preparation of polymeric systems sensitive to H_2_O_2_ through the cleavage of boronic ester compounds is a straightforward approach. Boronic ester compounds can be introduced to the motifs of polymeric NP’s design and the cleavage leads to polymer backbone degradation followed by cargo release. With these strategies small hydrophobic chemotherapeutic agents can be released upon exposure to biologically relevant oxidative conditions, e.g., from μM to mM concentrations of H_2_O_2_ [124,125,126]. Boronic ester based polyesters undergo oxidation with an insertion of oxygen to the linkage between boronic ester and the polyester chain or drug molecule of interest. In the presence of water this linkage undergoes hydrolysis resulting in the cleavage of the boronic ester linkage and subsequently polymer degradation or molecules release.

Several boronic ester polymer derivatives have been studied for ROS-responsivity among which aryl boronic esters with either ester or ether linkages show superior ROS-dependent degradation kinetics. This was demonstrated in one study designed by de Garcia Lux et al., where two new polymers were designed differing with respect to the linkage between the boronic ester group and the polymeric backbone: Either direct or via an ether linkage. The polymers were synthesized by polycondensation reaction using adipic acid as a co-monomer. NPs encapsulating a model of hydrophobic probe, Nile Red (NR), were formulated and the degradation kinetics of the polymer NPs was studied by monitoring the release of the model probe under varying concentrations of H_2_O_2_. The NPs formulated from the polymers of arylboronic esters with ether linkages have shown to be extremely sensitive to H_2_O_2_ at concentrations ~50 μM oppositely to the NPs formulated of arylboronic esters directly linked requiring about 1 mM H_2_O_2_ for the same amount of Nile Red released. The efficacy of the NPs to release the NR was probed upon incubation with activated neutrophils, simulating a physiologically relevant ROS-rich environment in vitro, where the NPs formulated from a polymer of arylboronic esters with ether linkage showed a two-fold enhancement of release if compared with the NPs with directly linked arylboronic esters while controls showed a nonspecific response to ROS producing cells [125]. In a similar approach, Jäger et al., prepared polymers containing phenylboronic esters and monomers bearing alkyne moieties for the attachment of fluorescent dyes via polycondensation reaction. The polymers were intended for the monitoring of the polymer fate in vitro. The polymers display triggered self-immolative degradation in the presence of ROS with the capability of cellular imaging. This was demonstrated after preparation of NPs encapsulating the fluorescent model probe NR (NR is fluorescent in the hydrophobic environment with quenched fluorescent in aqueous medium due to polarity changes in the micro-surrounding). The quenching of the NR probe was much higher in prostate cancer cells (PC-3 cells) compared to in Human fibroblast cells (HF cells). Moreover, the quenching of the NR was 2.5 times higher in PC-3 cells for the ROS-responsive polymer when compared with the NR quenched from one prepared polymer counterpart (without the phenylboronic ester monomer). These studies indicated that ROS-induced polymer degradation and NR release demonstrating the polymer’s potential for specifically release the cargo in ROS-rich intracellular environments. Further the authors covalently attached a fluorescent dye Alexa-Fluor 647 azide (Alx) by click-chemistry and monitored simultaneously the polymer degradation and the NR release in different cells by fluorescence life-time images (FLIM) and compared with the polymer counterpart. After 8 h incubation the NR and the Alx-bounded to the ROS-responsive polymer were co-localized to a high extent in HF cells whereas the cytoplasm was more homogeneously colored with the released NR observed in ROS-producing PC-3 cells. Finally the chemotherapeutic drug paclitaxel (PTX) was loaded, and enhanced in vitro cytotoxicity was observed for H_2_O_2_-responsive particles compared to the counterpart polymer-loaded to PTX (non-responsive to H_2_O_2_) in three different cancer (PC-3, HeLa and DLD1) cell lines [126].

#### 2.3.4. Polyoxalates

Oxalate based polyesters has been well known for being easily, quickly and specifically oxidized by H_2_O_2_ to form the corresponding alcohols and 1,2-dioxetanedione or other kinds of high energy intermediates, which can rapidly react with an appropriate fluorophore molecule to form an activated complex. After decomposition of the complex along with CO_2_ release, the excited fluorophore decays to the ground state with a fluorescent emission [127,128]. Thus, strategically placing aryloxalate ester bonds within polymeric NPs backbones is a particularly straightforward approach that can induce their degradation and release the cargo upon exposing to H_2_O_2_. In this way, polyoxalate polymers are usually prepared in one-pot synthetic pathway by polycondensation reaction. This strategy was explored for example by Lee et al., via the condensation reaction of oxalyl chloride with 4-hydroxybenzyl alcohol and 1,8-octanediol. Polyoxalate NPs were prepared by the oil/water (o/w) emulsion technique encapsulating different fluorescent dyes (perylene, rubrene or pentacene). Upon exposure to H_2_O_2_, the polyoxalate NPs degraded along with high energy intermediates production and subsequent dye emission showing high sensitivity to H_2_O_2_ in vitro over a linear range from 0.25 to 10 mM. The Pentacene-loaded polyoxalate NPs were capable of imaging hydrogen peroxide in the peritoneal cavity of mice during a lipopolysaccharide-induced inflammatory response in a mouse leg [127].

In a similar approach, Kang et al., designed H_2_O_2_-responsive copolymer NPs with strong antioxidant properties for therapeutic and diagnostic purposes based on the utilization of p-hydroxybenzyl alcohol (HBA), oxalyl chloride and 1,4-cyclohexanedimethanol. HBA is a major active pharmaceutical ingredient in *Gastrodia elata* Blume, which has been widely used as herbal agents for the treatment of oxidative stress-related diseases. These newly synthesized HBA polyoxalate NPs undergo complete H_2_O_2_-mediated degradation, releasing the active form of HBA and exerting its therapeutic effect demonstrated in in vitro experiments by using LPS-activated RAW 264.7 cells. The HBA based polyoxalate NPs demonstrated strong antioxidant and anti-inflammatory activities by inhibiting the production of nitric oxide and reducing TNF-*α* levels [129]. In forthcoming work, Lee et al., prepared polyoxalate NPs comprising oxalyl chloride, 1,4-cyclohexanedimethanol and vanillyl alcohol (VA) as backbones building blocks. VA is also an active pharmaceutical ingredient in *Gastrodia elata* Blume with probed antioxidant and anti-inflammatory properties. These polyoxalate NPs containing VA were designed to degrade and release VA, which is able to reduce the generation of ROS, and exert anti-inflammatory and anti-apoptotic activity. The polyoxalate vanillyl NPs specifically reacted with overproduced H_2_O_2_ and exerted highly potent anti-inflammatory and anti-apoptotic activities that reduced cellular damages as demonstrated in models of hind-limb ischemia/reperfusion injuries in vivo [130]. By taking advantage of such a strategy, Jäger et al. re-utilized a chemotherapeutic drug withdrawn from market due to dose-related adverse effects. Via a one-pot condensation reaction new ROS-sensitive, self-immolative biodegradable polyoxalate prodrug based on the anticancer chemotherapeutic hormone analog diethylstilbestrol was synthesized. The NPs prepared from the diethylstilbestrol prodrug undergoes self-immolative degradation releasing diethylstilbestrol in ROS-rich niches, e.g., in the cancer cells. This new ROS self-immolative diethylstilbestrol polymeric based prodrug circumvents the necessity of a linker between the polymeric chain and the chemotherapeutic drug, exhibiting more specific drug release and minimum adverse effects to non-ROS overproducing cells as demonstrated in in vitro experiments. Altogether these new NPs based on the diethylstilbestrol polymeric prodrug reduced the adverse effects of an effective and largely applied chemotherapeutic drug applied for hormone-dependent cancers [131].

Despite the use of oxalates as main building blocks agents, alternatively oxalate groups were also employed as pendant phenyl alkyl hybrid oxalate groups in a polynorborene-type copolymer with oligo(ethylene glycol) chains synthesized by ring-opening metathesis polymerization for the preparation of ROS-sensitive micelles [132]. They were also used as block linkages between a PLGA and a poly(ethylene glycol block) in a AB-type amphiphilic ROS-triggered nanoparticle-based antigen delivery system with pronounced vaccine-induced immune responses in vivo [133]. Recently, they increase the degradation kinetics profile of polycaprolactone. In the latter case, oxalyl chloride was reacted with *α,ω*-dihydroxy oligocaprolactone to generate an oligocaprolactone multiblock copolymer-oxalate. Microspheres prepared form the new PCL-oxalate copolymer were completely cleared two months after in vivo implantation in the mice model demonstrating similar degradation rate as the PLGA microspheres prepared as controls [134].

### 2.4. Enzymatically Labile Polyesters

The aliphatic polyesters excel in the biodegradation processes among the polymers. It is given by the nature of the chemical structure because the ester bond undergoes a hydrolysis reaction readily. The rate of the process is very slow in the mild condition but it rises up when the temperature increases, the pH of solution deviates from neutral or, especially, when certain enzymes—the esterases—are present in the solution. This chemical process can cause troubles in material applications in respect of the loss of mechanical properties. However, when considering application as stimuli-responsive polymers where the degradation process is desirable, the hydrolysis can play a crucial role in the system. The polymeric materials for drug delivery applications are deliberately designed from degradable polymers so the encapsulated drug is released with the respect of the mass loss given by the rate of degradation [135,136].

The rate of the enzymatic hydrolysis can be tuned by the chemical structure and physical properties of the polyester. In addition, the polymer structure is often intentionally designed and synthetized in the way of faster degradability for these applications. The degradability of aliphatic polyesters generally increases with a lower content of the crystalline structure and it can be done by the introduction of irregularities to the polymeric structure e.g., by the presence of the side groups [137].

The very important factor in the enzymatic degradation is the enzyme itself. It is well known that the enzymes serve as selective biocatalysts in living systems. Many enzymes became very substrate-selective during evolution (e.g., polymerases). On the other hand, there are enzymes that do not exhibit such strict selectivity [138]. The digestive enzymes break down the macromolecules in the nutrients so the need of accurate scission is not essential as it is in the case of reproduction. In addition, there are several digestive enzymes that have the same effect but the kinetics changes depending on the nature of the substrate. It was already demonstrated in many publications, e.g., the rate of enzymatic degradation of polyester backbone in poly(ethylene oxide)-block-poly(*ω*-caprolactone) was significantly faster when the bacterial lipase from *Pseudomonas* sp. was used compared to the porcine pancreatic lipase [139].

The rate would differ in different environments, in medical terms—in different biological compartments. Therefore, the polyester-based polymers have become very popular in the smart drug delivery systems. The elevated concentration of hydrolases with the combination of lowered pH in some biological compartments such as, for example, in endosomes or lysosomes would cause an increased degradation rate of ester bond. The rate of the enzymatic hydrolysis is very individual because of the enzyme specificity, as mentioned above [140,141].

In the study of Trousil et al., the self-assembled nanoparticles based on poly(ethylene oxide)-block-poly(*ω*-caprolactone) were loaded with rifampicin, a first-line antituberculotic agent and cornerstone of modern antituberculotic therapy. It was shown that the degradation rate depends not only on the nature of the substrate but also on the blocks’ sizes and the final nanoparticles’ sizes and their distribution, suggesting that the nanobead-based intervention’s trafficking is a complex phenomenon. Despite this fact, some trends in the relationships have been seen. What is more, the antituberculotic effect has been demonstrated both in vitro and in vivo. Using a well-established in vitro model of infected macrophage-like cells, it was found that the polyester-based nanoparticles are safe and able to suppress the virulent mycobacterial strain H37Rv-caused infect. Additionally, *Mycobacterium marinum*-infected zebrafish embryos were used to assess the treatment effect in vivo. The zebrafish larvae treated by the rifampicin-loaded nanobead-based intervention have shown higher survival values in Figure 9. Given these results, it is believed that these kinds of nanoparticles have a potential for the treatment of intracellularly persisting bacteria-caused infections such as tuberculosis due to passive targeting to macrophage-like cells. The enzymatic degradation of the polyester part of nanoparticles causes the drug release in the endosomes so the drug gets closer and at a higher concentration at the site of the mycobacteria-caused infection. Hence, the treatment is more efficient [142].

Altogether the potential application of the selectively biodegradable polyesters mentioned along the manuscript is summarized in Table 2.

## 3. Conclusions

Degradable polyesters have been subject of great attention along decades because of their nature characteristics and because they can be degraded into smaller, biocompatible molecules that are easily cleared via conventional paths, such as renal filtration and hepatic metabolization. With the advances in polyester synthesis, several polymers were built-up with varied morphologies, stereocomplexations, chemical compositions and with different degradation profiles. Major progress in the field of degradable polyesters was achieved tailoring the polymers to be responsive to changes in physiological conditions. In this review we presented the state-of-the-art of the synthesis, structure-properties, degradation characteristics and biomedical applications of biologically responsive polyesters. Polyesters responsive to enzymes, to the changes on pH, reductive degradable polyesters and reactive oxygen species degraded polyesters were discussed in detail. These new stimuli-responsive polyesters were employed as biomaterials in several fields such as in surgery, tissue repair and regeneration, tissue engineering and sustained drug delivery systems for various kinds of bioactive molecules demonstrating their broad applicability, their success and the generally bright future of the field of stimuli-responsive polyester. However, progresses in the field are still necessary with more studies and methods for a better understanding of polyester degradation characteristics as well as for the development of novel degradable polyester devices for actual medical and pharmaceutical application challenges. With the enormous possibility of monomer selection and polymerization processes a multitude of polyesters can be designed with control over degradation, biocompatibility and physical properties to obtain diverse materials for biomedical applications. Furthermore, novel materials with several different responsiveness’ can be created by the combination of different monomers, which holds promises for several different applications. As demonstrated just from one small sort of responsive polyesters described herein, they offer a facile route to a plethora of materials with a variety of thermal, degradation and mechanical properties. Altogether, the development and production of synthetic selectively degradable polymers for applications as biomaterials is of the upmost importance for the advancement of effective biomedical therapeutics and the future of the field.

## Figures and Tables

**Figure 1 polymers-11-01061-f001:**
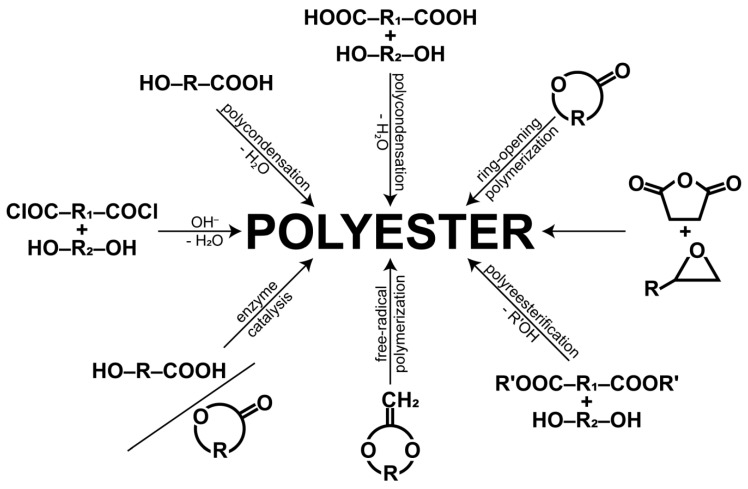
An overview of the synthetic routes used for the synthesis of polyesters.

**Figure 2 polymers-11-01061-f002:**
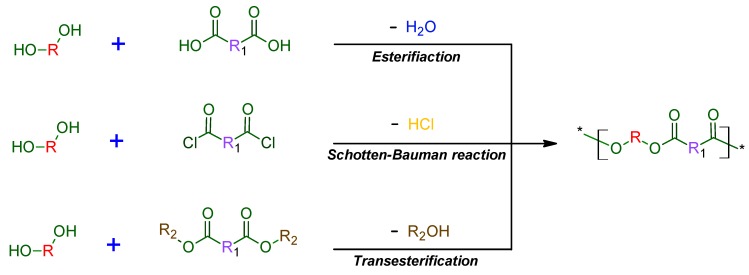
Examples of stepwise polycondensation reactions for the preparation of polyesters [10].

**Figure 3 polymers-11-01061-f003:**
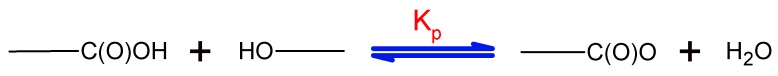
The equilibrium constant of polycondensation reaction of carboxylic acids with alcohols [10].

**Figure 4 polymers-11-01061-f004:**
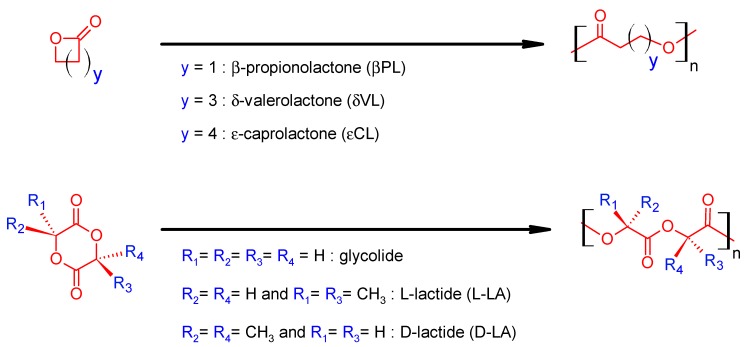
Monomers of choice for ring opening polymerization (ROP) polymerization [10].

**Figure 5 polymers-11-01061-f005:**
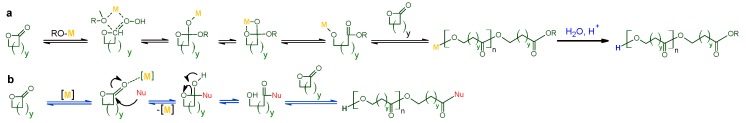
Mechanism of ROP of lactones using organometallics [M] as (**a**) an initiator in the “coordination-insertion” mechanism and (**b**) as a catalyst in the presence of nucleophiles (Nu) [10].

**Figure 6 polymers-11-01061-f006:**
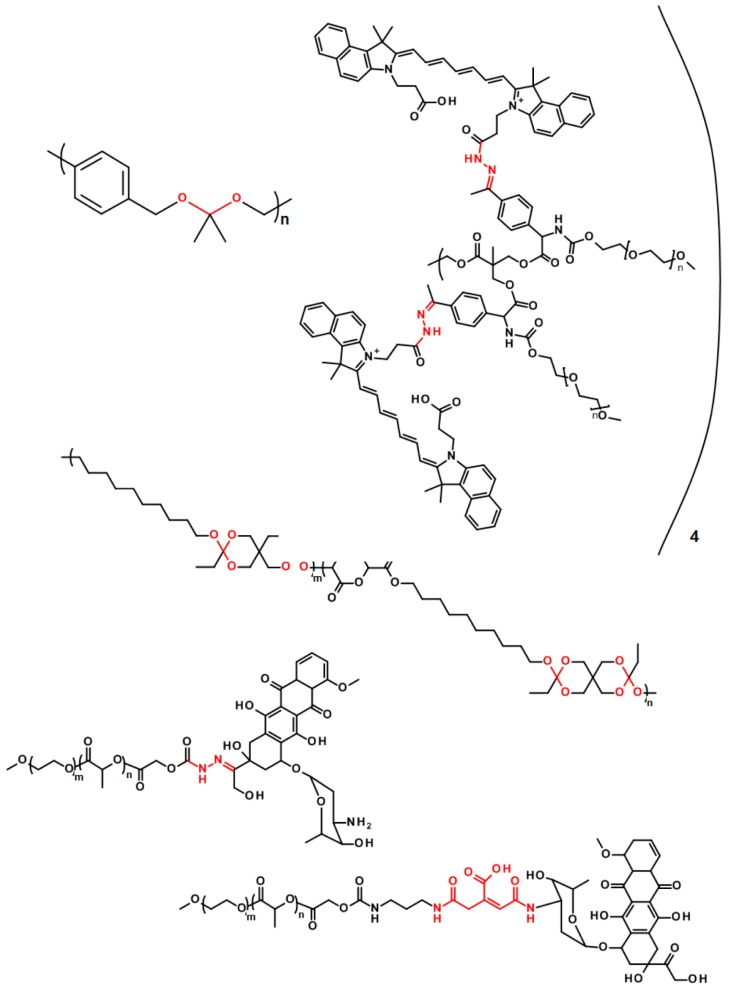
Polymer structures with pH-sensitive linkages.

**Figure 7 polymers-11-01061-f007:**
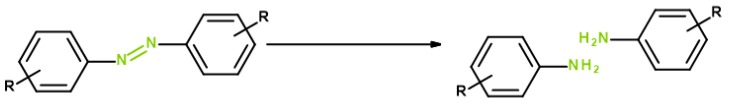
The reductive cleavage of azo-dye linkage.

**Figure 8 polymers-11-01061-f008:**
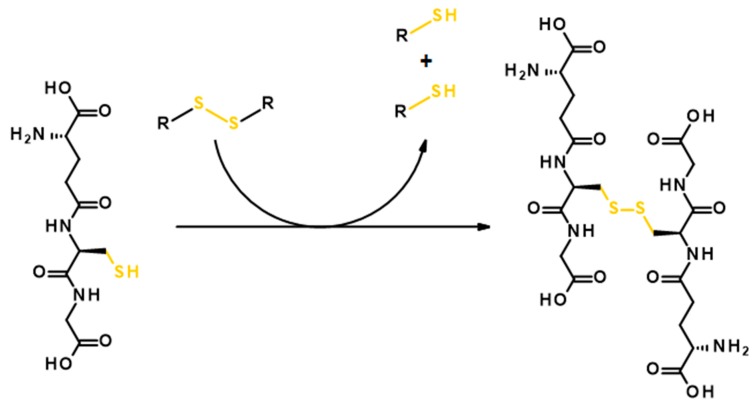
The scheme of reductive degradation of the disulfide bond with glutathione.

**Figure 9 polymers-11-01061-f009:**
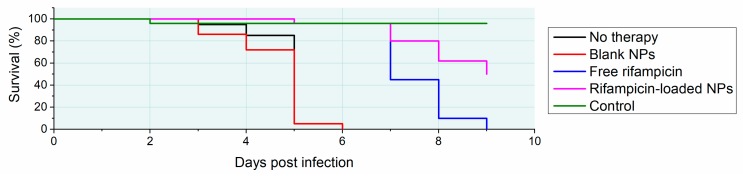
In vivo testing of antituberculotic nanobead-based intervention. Zebrafish larvae were infected with *Mycobacterium marinum* and treated with free rifampicin, rifampicin-loaded nanoparticles and blank nanoparticles at a dose of 10 mg/kg. Cumulative mortality is shown. Modified based on data from reference [142].

**Table 1 polymers-11-01061-t001:** Representative oxidation-responsive polyesters and their oxidation products.

ROS-Sensitive Materials	Chemical Structure and Oxidation	References
Poly(propylene sulfide)s	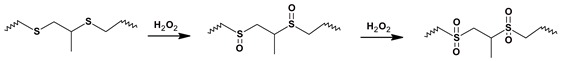	[109,110,111,112,113,114,115,116,117,118,119]
Selenium containing polyesters	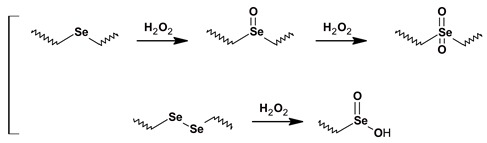	[120,121,122,123,124]
Arylboronic ester containing polyesters	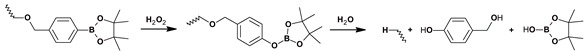	[125,126,127]
Polyoxalates	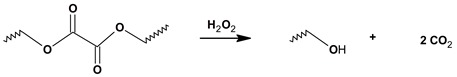	[128,129,130,131,132,133,134,135]

**Table 2 polymers-11-01061-t002:** Selectively biodegradable polyesters and their biomedical applications.

Polyester	Sensitive Linkage	Biomedical Application	References
pH-sensitive	Ketal	Anti-inflammatory drug (Dexamethasone);	[80]
	Orthoester	Injectable drug release device, Cell scaffold;	[77]
	Cis-aconityl	Chemotherapy (Doxorubicin);	[79]
Reductive-labile	Azo groups	Colon therapy (Imaging);	[86]
	Disulfide	Chemotherapy (Paclitaxel);	[99]
	Propylene sulfide	Chemotherapy (Doxorubicin);	[109]
	Propylene sulfide	Vaccination (antigens);	[105]
	Propylene sulfide	Chemotherapy (Paclitaxel);	[107]
ROS-labile	Selenide, diselenide	Chemotherapy (Doxorubicin);	[112,113]
	Aryl boronic esters	Chemotherapy (Paclitaxel);	[118]
	Oxalate	Chemotherapy (Dyethylstilbestrol);	[123]
	Oxalate	Antioxidant and anti-inflammatory (Vanilyl alcohol);	[122]
Enzymatically-labile	Ester bond	Antituberculotic (Rifampicin);	[134]

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
