# Peer review of "Selectively Biodegradable Polyesters: Nature-Inspired Construction Materials for Future Biomedical Applications"

_polymers, 2019, doi:10.3390/polym11061061_

Round 1
Reviewer 1 Report
1. In this review article, the author introduces the smart drug delivery systems based on biodegradable polyesters. It also describes synthetic method of polyesters and their response to the environment, such as pH-labile polyesters, reductively labile polyesters, reactive oxygen species (ROS)-labile polyesters, and enzymatic labile polyesters. This is a rich, interesting and useful review article.
2. There are many typos and two period errors in the article, please correct. (For example: polyesters. ., strictly obeyed.., polycondensation.., Figure 66...etc. )
3. In the article, a number of polyester nanoparticles are used for in vitro and in vivo research applications, however the safety data of nanoparticles is lacking. It is recommended to supplement the safety test data of each research.
4. The study of intestinal microbiota has become more and more popular in recent years. In the reductively labile polyesters section, poly(ether-ester)s can be degraded by the azoreductase of bacteria in the intestine, therefore it can be used for colon-specific drug release. However, its reference is 1995, I suggest that it can add research from more recent references.
5. In the polyester nanoparticles are used for in vitro and in vivo research applications, it is suggested that the efficacy of the drug embedding by different nanoparticles and the rate of promoting release can be explained. Rather than just introducing polyester nanoparticles to improve drug delivery in vitro or in vivo.
Author Response
Reviewer 1
1. In this review article, the author introduces the smart drug delivery systems based on biodegradable polyesters. It also describes synthetic method of polyesters and their response to the environment, such as pH-labile polyesters, reductively labile polyesters, reactive oxygen species (ROS)-labile polyesters, and enzymatic labile polyesters. This is a rich, interesting and useful review article.
Response: We thank the referee for the positive comment.
2. There are many typos and two period errors in the article, please correct. (For example: polyesters. ., strictly obeyed.., polycondensation.., Figure 66...etc. )
Response: These are typos that were corrected. We thank to the reviewer for pointed out. The whole manuscript was carefully revised accordingly.
3. In the article, a number of polyester nanoparticles are used for in vitro and in vivo research applications, however the safety data of nanoparticles is lacking. It is recommended to supplement the safety test data of each research.
Response: The safety of polyester nanoparticles for in vitro and in vivo applications was discussed more in detail along the manuscript including referring to the original papers describing the particular systems as primary source. These cited papers contain safety data if existent for the particular system described.
4. The study of intestinal microbiota has become more and more popular in recent years. In the reductively labile polyesters section, poly(ether-ester)s can be degraded by the azoreductase of bacteria in the intestine, therefore it can be used for colon-specific drug release. However, its reference is 1995, I suggest that it can add research from more recent references.
Response: Two recent reviews were added to the manuscript covering the field of polyester and colon-specific release.
REF 88. Lu, L.; Chen, G.; Qiu, Y; Li, M.; Liu, D.; Hu, D.; Gu, X.; Xiao, Z. Nanoparticle-based oral delivery systems for colon targeting: principles and design strategies. Science Bulletin 2016, 61, 670-681.
REF 89. Rajpurohit, H.; Sharma, P.; Sharma, S.; Bhandari, A.; Polymers for Colon Targeted Drug Delivery. Indian J Pharm Sci., 2010, 72, 689-696.
5. In the polyester nanoparticles are used for in vitro and in vivo research applications, it is suggested that the efficacy of the drug embedding by different nanoparticles and the rate of promoting release can be explained. Rather than just introducing polyester nanoparticles to improve drug delivery in vitro or in vivo.
Response: We thank the referee for the comment. One paragraph explaining the polyester drug interactions and drug release was added to the revised version of the manuscript (highlighted). Stimuli-dependent degradation-based release of active components is already discussed elsewhere in the manuscript.
Page 6, lines 166 to 190
1.2. Drug release from polyesters in general
The potential of protect the active drug component from degradation together with the sustaining of their release, as well the capability of modulate the active drug diffusion and polymer degradation resulting in inert body-friendly degradation byproducts, make polyesters successful candidates for drug delivery systems in biomedical applications. The encapsulation of drugs into polyester-based nanosystems, subsequent drug release and to some extent also polymer degradation are generally dependent on the polymer/matrix interaction with the dissolved (compound at the amorphous state) or dispersed (compound at crystalline state) drug.[1-3] This is directly related to the solubility (for the crystalline host molecules) and/or miscibility (for the amorphous load) of the drug with the polymer matrix and vice-versa. The polymer crystallinity can also play important rule for the encapsulation, drug release and degradation. Some studies reported that the drug loading is decreased with the increasing of the polymer crystallinity and glass transition temperature (Tg),[3-6] other studies reported that the drug loading capacity for polyester nanoparticles can be significantly enhanced by hydrophobic effects between the polymer and the drug and in combination with hydrogen bonding, electrostatic interaction and dipole-dipole interactions modifying the drug release profile.[7-10] In addition, the degradation behavior of the polyester matrix is an important prerequisite for the potential drug release. There are two main mechanisms involved on the degradation of polyesters and they are dependent on the relative rates of water diffusion into the polymer matrix and on the polymer degradation rate.[1,2] In the case where the rate of polymer degradation is faster than the rate of water diffusion into the polymer matrix the mechanism is called surface degradation. On the contrary, when diffusion of water into the matrix is faster than polymer degradation and the whole matrix is affected by degradation and erosion, the process is called bulk degradation. In general, under biological conditions (in vitro and in vivo) the degradation of polyesters without specific responsive linkage (described in detail hereafter) proceeds by random hydrolytic cleavage of ester linkages.
Reviewer 2 Report
Tomáš Urbánek et al have written a review entitled “Selectively Biodegradable Polyesters: Nature-Inspired Construction Materials for Future Biomedical Applications”. The authors do a good job of covering the current literature on this growing research field which dates back to, as the authors mention in the abstract, the last half century. It does a good job in selecting out a subset of the field which is becoming more important to future biomaterials. But regardless of how comprehensive the review may appear, there is a great drawback in its current state: it does not read well throughout. There are sentences that have multiple grammar mistakes and some are simply unfinished and confusing. I will list some examples of some of the poorly written sentences here however these are simply representative of many more that occur throughout.
Page 4 line 108: “Mostly because ROP processes several advantages compared to traditional condensation polymerization, e.g., high conversion without necessity of removal of reaction byproducts, shorter reaction times, mild synthetic conditions, , and the use of stoichiometric balance of monomers and sometimes proceeds in a “living” manner, without side reactions which allows good control of the polymer characteristics (narrow molecular weight distribution and predictable molecular weight ).[6,10,22]”
Page 6 line 158: “Moreover, their copolymerization with fatty acids such as dilinoleic acid, which are naturally occurring body compounds,[58,59] allow the preparation of more hydrophobic biodegradable polymers suitable for fine-tuned hydrophobic drugs encapsulation via hydrophobic interactions when they are used as drug nanocarriers.[8,9,53,60,61]”
Both of these sentences are too long, wordy, confusing and have no structure. The first example appears unfinished.
The following sentences make no sense at all as they are written:
Page 11, line 334: “The NPs prepared from the newly synthesised polyester containing selenide group respond much faster to H2O2 than NPs prepared by the same pathway, however, containing thioether groups most likely to the superior sensitivity of the selenide toward H2O2.[114]” ??
Page 12, line 380 “These studies indicated that ROS-induced degradation demonstrating the polymer’s potential to specifically trigger the cargo release in an ROS-rich intracellular environment.:” ??
Page 13, line 424: “By taking advantage of these strategy, Jäger et al., take advantage of this strategy to enable the re-utilization of a chemotherapeutic drug marketed withdrawn due to dose-related adverse effects.” ???
Page 14, line 466: “In the time of the evolution, some enzymes have become very selective even with supporting error detection and elimination, for example in the case of polymerases” ??
Other issues which need to be addressed:
-The central topic of the review is a subset of the polyester field, namely, “Biodegradable polyester materials containing external stimuli-sensitive linkages”. It is therefore appropriate that the introduction quickly diverges from a discussion on synthesis and mechanisms of polyesters. However, this transition is awkward in the review as it is written, it should flow better. I think it would be advantageous to insert some subheadings under the introduction (such as “synthesis and mechanisms”). Section 2 should have a heading of “Polyesters with stimuli-sensitive linkages” and then each of the sections could be subheadings, such as 2.1 pH-labile polyesters.
- All of the figures have different font sizes, sometimes even within one figure itself (such as figure 1). For figures that are taken from publications I think it’s acceptable to reproduce them for the purposes of a review. As long as the author seeks approval from the journal publisher.
-Section 4 (ROS-labile polyesters) needs more figures throughout. Also the subheadings needs numbers.
- One of the subsections of section 4, is “polysulfides”. I disagree that the polymers that the author is referring to are polysulfides as none of the structures described have structures with multiple, consecutive repeating units of sulfur, eg R-SSn-R, where n>1. I suggest that the authors use the same naming classification as was used by one of the reviews that they refer to (101. Song, C.-C.; Du, F.-S.; Li, Z.-C. Oxidation-responsive polymers for biomedical applications. Journal of Materials Chemistry B 2014, 2, 3413-3426.), i.e, use Poly(propylene sulfide)s, not polysulfides.
- There are several instances that the authors mention the word ‘targeted’ with respect to the site where the nanoparticles are expected to degrade the most e.g.:
Page 6 line 166 :“Typically, the cancer and inflamed tissues are targeted in this way (in the case of inflammation, the presence of reactive oxygen species is also used for targeting, see below)[62]”
Page 8 line 208 “They utilize i) the reduction of disulfide bond by glutathione targeting cancer cells and ii) the bacterial reduction of aromatic azo-dye which allows colon-selective degradation.[82,83]”
Page 8, line 222: “These polyesters containing the azo linkages can be used in colon targeting therapy.[87]”
Page 9, line 230: “Considerable number of systems has been invented for drug targeting to the cancer tissue with the use of the disulfide bond”
Page 9 line 244: “This phenomenon can be taken in advantage in drug design and many researchers have suggested polyester systems containing disulfide bond as a drug carrier with the target of cancer cells.”
Page 10 , line 263:”The importance of ROS that it’s involved in several pathological sates and cellular signalization have attracted great attention towards the development of chemical tolls to ROS-target specific niches, and ROS-responsive micro-or nano-delivery systems.”
Page 13, line 428:”The NPs prepared from the diethylstilbestrol prodrug undergoes self-immolative degradation releasing diethylstilbestrol in ROS-rich niches, e.g., in the targeted cancer cells.”
In all these instances it’s more about a greater selectivity for the NP to cleave at tumor cells, more so than a classic ‘targeted’ approach used and described for cancers. The latter is normally a NP that includes, as part of its structure, a targeting ligand or antibody so that it can be retained and taken up by the tumour cells. I think the best way to overcome this problem is to either use a different word than ‘targeting’ or to explain early on what the authors mean by this. To avoid confusion.
Author Response
Reviewer 2
Tomáš Urbánek et al have written a review entitled “Selectively Biodegradable Polyesters: Nature-Inspired Construction Materials for Future Biomedical Applications”. The authors do a good job of covering the current literature on this growing research field which dates back to, as the authors mention in the abstract, the last half century. It does a good job in selecting out a subset of the field which is becoming more important to future biomaterials. But regardless of how comprehensive the review may appear, there is a great drawback in its current state: it does not read well throughout. There are sentences that have multiple grammar mistakes and some are simply unfinished and confusing. I will list some examples of some of the poorly written sentences here however these are simply representative of many more that occur throughout.
Page 4 line 108: “Mostly because ROP processes several advantages compared to traditional condensation polymerization, e.g., high conversion without necessity of removal of reaction byproducts, shorter reaction times, mild synthetic conditions, , and the use of stoichiometric balance of monomers and sometimes proceeds in a “living” manner, without side reactions which allows good control of the polymer characteristics (narrow molecular weight distribution and predictable molecular weight ).[6,10,22]”
Page 6 line 158: “Moreover, their copolymerization with fatty acids such as dilinoleic acid, which are naturally occurring body compounds,[58,59] allow the preparation of more hydrophobic biodegradable polymers suitable for fine-tuned hydrophobic drugs encapsulation via hydrophobic interactions when they are used as drug nanocarriers.[8,9,53,60,61]”
Both of these sentences are too long, wordy, confusing and have no structure. The first example appears unfinished.
Response: We thank for the reviewer comment. These sentences were revised and corrected (highlighted). All the manuscript was carefully revised accordingly.
Page 4, line 108: “The ROP polymerization processes several advantages compared to traditional condensation polymerization, e.g., high conversion without the necessity of removal of reaction byproducts, shorter reaction times, mild synthetic conditions, and the use of stoichiometric balance of monomers. Moreover, the ROP polymerization sometimes proceeds in a “living” manner, without side reactions, which allows good control of the polymer characteristics (narrow molecular weight distribution and predictable final molecular weight of polymer).[6,10,22]”
Page 6, line 158: “Moreover, the copolymerization of alkylene succinates with fatty acids (naturally occurring body compounds) [58,59], such as dilinoleic acid, allow the preparation of more hydrophobic biodegradable polymers. This more hydrophobic polymers are suitable for fine-tune hydrophobic drugs encapsulation via polymer-drug interactions when, for example, this polymers are used as drug nanocarriers.[8,9,53,60,61]”
The following sentences make no sense at all as they are written:
Page 11, line 334: “The NPs prepared from the newly synthesised polyester containing selenide group respond much faster to H2O2 than NPs prepared by the same pathway, however, containing thioether groups most likely to the superior sensitivity of the selenide toward H2O2.[114]” ??
Response: We thank for the reviewer comment. This sentence was revised and rewritten (highlighted).
“The NPs prepared from this newly synthesized polyester containing pendant selenide groups respond much faster to H2O2 than NPs prepared by the same pathway (ROP of caprolactone), however containing thioether as pendant groups, most likely due to the superior sensitivity of the selenide pendant groups towards H2O2.[114]“
Page 12, line 380 “These studies indicated that ROS-induced degradation demonstrating the polymer’s potential to specifically trigger the cargo release in an ROS-rich intracellular environment.:” ??
Response: We thank for the reviewer comment. This sentence was revised and rewritten (highlighted).
“These studies indicated that ROS-induced polymer degradation and NR release demonstrating the polymer’s potential for specifically release the cargo in ROS-rich intracellular environments“
Page 13, line 424: “By taking advantage of these strategy, Jäger et al., take advantage of this strategy to enable the re-utilization of a chemotherapeutic drug marketed withdrawn due to dose-related adverse effects.” ???
Page 13, line 424: “By taking advantage of such strategy, Jäger et al. re-utilizated a chemotherapeutic drug withdrawn from market due to dose-related adverse effects.“
Page 14, line 466: “In the time of the evolution, some enzymes have become very selective even with supporting error detection and elimination, for example in the case of polymerases” ??
Page 14, line 466: Many enzymes became very substrate-selective during evolution (e.g., polymerases).
Other issues which need to be addressed:
-The central topic of the review is a subset of the polyester field, namely, “Biodegradable polyester materials containing external stimuli-sensitive linkages”. It is therefore appropriate that the introduction quickly diverges from a discussion on synthesis and mechanisms of polyesters. However, this transition is awkward in the review as it is written, it should flow better. I think it would be advantageous to insert some subheadings under the introduction (such as “synthesis and mechanisms”). Section 2 should have a heading of “Polyesters with stimuli-sensitive linkages” and then each of the sections could be subheadings, such as 2.1 pH-labile polyesters.
Response: We thank the reviewer for the suggestion. Headings and subheading are given now in the revised version of the manuscript (highlighted).
- All of the figures have different font sizes, sometimes even within one figure itself (such as figure 1). For figures that are taken from publications I think it’s acceptable to reproduce them for the purposes of a review. As long as the author seeks approval from the journal publisher.
Response: The figure 1 of the manuscript was corrected accordingly. We thank the reviewer for the suggestion.
-Section 4 (ROS-labile polyesters) needs more figures throughout. Also the subheadings needs numbers.
Response: We thank the reviewer for the suggestion. Subheadings are given now in the revised version of the manuscript (highlighted). As there are already a number of pictures in the whole manuscript, we kept the actual number to save the journal space.
- One of the subsections of section 4, is “polysulfides”. I disagree that the polymers that the author is referring to are polysulfides as none of the structures described have structures with multiple, consecutive repeating units of sulfur, eg R-SSn-R, where n>1. I suggest that the authors use the same naming classification as was used by one of the reviews that they refer to (101. Song, C.-C.; Du, F.-S.; Li, Z.-C. Oxidation-responsive polymers for biomedical applications. Journal of Materials Chemistry B 2014, 2, 3413-3426.), i.e, use Poly(propylene sulfide)s, not polysulfides.
Response: We thank the reviewer for the suggestion. In the revised version of the manuscript the term polysulfide’s were substituted by Poly(propylene sulfide)s (highlighted).
- There are several instances that the authors mention the word ‘targeted’ with respect to the site where the nanoparticles are expected to degrade the most e.g.:
Page 6 line 166 :“Typically, the cancer and inflamed tissues are targeted in this way (in the case of inflammation, the presence of reactive oxygen species is also used for targeting, see below)[62]”
Page 6, line 166: “Typically, the polymer degradation may be selective in this way in cancer and inflamed tissues (in the case of inflammation, the presence of reactive oxygen species is also used for specific drug release or polymer degradation, see below) [62].”
Page 8 line 208 “They utilize i) the reduction of disulfide bond by glutathione targeting cancer cells and ii) the bacterial reduction of aromatic azo-dye which allows colon-selective degradation.[82,83]”
Page 8, line 208: “They utilize i) the reduction of disulfide bond by glutathione specifically in cancer cells and ii) the bacterial reduction of aromatic azo-dye which allows colon-selective degradation.[82,83]”
Page 8, line 222: “These polyesters containing the azo linkages can be used in colon targeting therapy.[87]”
Page 8, line 222: “These polyesters containing the azo linkages can be used in colon-specific therapy.[87]”
Page 9, line 230: “Considerable number of systems has been invented for drug targeting to the cancer tissue with the use of the disulfide bond”
Page 9, line 230: “Considerable number of systems has been invented for drug specific release at the cancer tissue with the use of the disulfide bond”
Page 9 line 244: “This phenomenon can be taken in advantage in drug design and many researchers have suggested polyester systems containing disulfide bond as a drug carrier with the target of cancer cells.”
Page 9, line 244: “This phenomenon can be taken in advantage in drug design and many researchers have suggested polyester systems containing disulfide bond as a drug carrier aiming specific release or polymer degradation on cancer cells.”
Page 10 , line 263:”The importance of ROS that it’s involved in several pathological sates and cellular signalization have attracted great attention towards the development of chemical tolls to ROS-target specific niches, and ROS-responsive micro-or nano-delivery systems.”
Page 10, line 263: ”The importance of ROS that it’s involved in several pathological sates and cellular signalization have attracted great attention towards the development of chemical tools for specific delivery to ROS-rich niches, and ROS-responsive micro-or nano-delivery systems.”
Page 13, line 428:”The NPs prepared from the diethylstilbestrol prodrug undergoes self-immolative degradation releasing diethylstilbestrol in ROS-rich niches, e.g., in the targeted cancer cells.”
Page 13, line 428: ”The NPs prepared from the diethylstilbestrol prodrug undergoes self-immolative degradation releasing diethylstilbestrol in ROS-rich niches, e.g., in the cancer cells.”
In all these instances it’s more about a greater selectivity for the NP to cleave at tumor cells, more so than a classic ‘targeted’ approach used and described for cancers. The latter is normally a NP that includes, as part of its structure, a targeting ligand or antibody so that it can be retained and taken up by the tumour cells. I think the best way to overcome this problem is to either use a different word than ‘targeting’ or to explain early on what the authors mean by this. To avoid confusion.
Response: We share the same opinion of the meaning “targeting” with the reviewer. In the revised version of the manuscript the term targeting is substituted by the terms: specific release, chosen environment, aiming the degradation and release etc.,). We thank the referee for the suggestion which was adopted.
Reviewer 3 Report
The polymeric nanomaterials have been extensively investigated for different applications especially in the biomedical fields. In this review, the authors summarized the recent development and advances about the synthesis of degradable polymers and their potential biomedical applications for tissue engineering, temporary implants, wound healing and drug delivery etc were involved. This topic should be of great interest and broad audience. The whole structure of this manuscript is good. I think this manuscript is publishable after the following issues could be well addressed. My detailed comments on this manuscript are listed below.
1. In this work, the synthesis of degradable polymers has been well addressed, however, the contexts about potential biomedical applications of these degradable polymers are not enough. I recommond the authors add some contexts about the biomedical applications of these polymers.
2. The synthesis of functional degradable polymers should be very interest, in this manuscript, the authors summarized the preparation of stimuli responsive degrade polymers for biomedical applications. The preparation of luminescent degrade polymers are recently attracted increase interest for biomedical applications. I suggest the authors add some context about the synthesis of fluorescent polymers espcially degrade polymers. Some related references (e.g. Journal of Colloid and Interface Science 508, 248-253, Journal of the Taiwan Institute of Chemical Engineers, 95, 234-240, Applied Materials Today 9, 145-160, Chemical Engineering Journal 337, 82-89, Journal of colloid and interface science 513, 198-204,Nanoscale 8 (38), 16819-16840, Nanoscale 7 (27), 11486-11508, Polymer Chemistry 5 (2), 356-360, Polymer Chemistry 5 (2), 399-404, Materials Science and Engineering: C 91, 201-207, Materials Science and Engineering: C 81, 416-421, Materials Science and Engineering: C 80, 708-714, Materials Science and Engineering: C 80, 578-583, Materials Science and Engineering: C 80, 411-416, Materials Science and Engineering: C 79, 563-569, Materials Science and Engineering: C 79, 590-595, Materials Science and Engineering: C 78, 862-867) should also be mentioned and cited in this place.
3. More detailed description about the cancer treatment using biodegradable polymers should be made in the revised manuscript.
4. The challenges and further directions about this field should be in-depth discussed during revisions. It is very important for a review manuscript.
Author Response
Reviewer 3
The polymeric nanomaterials have been extensively investigated for different applications especially in the biomedical fields. In this review, the authors summarized the recent development and advances about the synthesis of degradable polymers and their potential biomedical applications for tissue engineering, temporary implants, wound healing and drug delivery etc were involved. This topic should be of great interest and broad audience. The whole structure of this manuscript is good. I think this manuscript is publishable after the following issues could be well addressed. My detailed comments on this manuscript are listed below.
1. In this work, the synthesis of degradable polymers has been well addressed, however, the contexts about potential biomedical applications of these degradable polymers are not enough. I recommond the authors add some contexts about the biomedical applications of these polymers.
Response: We thank for the reviewer suggestion. The potential biomedical applications of these degradable polymers are now described at the revised version of the manuscript as Table 2 (highlighted).
Table 2. Selectively biodegradable polyesters and their biomedical applications.
2. The synthesis of functional degradable polymers should be very interest, in this manuscript, the authors summarized the preparation of stimuli responsive degrade polymers for biomedical applications. The preparation of luminescent degrade polymers are recently attracted increase interest for biomedical applications. I suggest the authors add some context about the synthesis of fluorescent polymers espcially degrade polymers. Some related references (e.g. Journal of Colloid and Interface Science 508, 248-253, Journal of the Taiwan Institute of Chemical Engineers, 95, 234-240, Applied Materials Today 9, 145-160, Chemical Engineering Journal 337, 82-89, Journal of colloid and interface science 513, 198-204,Nanoscale 8 (38), 16819-16840, Nanoscale 7 (27), 11486-11508, Polymer Chemistry 5 (2), 356-360, Polymer Chemistry 5 (2), 399-404, Materials Science and Engineering: C 91, 201-207, Materials Science and Engineering: C 81, 416-421, Materials Science and Engineering: C 80, 708-714, Materials Science and Engineering: C 80, 578-583, Materials Science and Engineering: C 80, 411-416, Materials Science and Engineering: C 79, 563-569, Materials Science and Engineering: C 79, 590-595, Materials Science and Engineering: C 78, 862-867) should also be mentioned and cited in this place.
Response: We thank for the reviewer suggestion. We carefully checked the references and we do not find these articles to be more important for the present manuscript than many other articles published about polyesters and stimuli responsive during the last years (searching on PubMed for “polyester AND responsive” showed more than 400 hits). The suggested papers from the referee are related more generally to polyurethanes, poly(amino acid)s or cyclodextrins and not directly to polyesters, the main scope of our review.
3. More detailed description about the cancer treatment using biodegradable polymers should be made in the revised manuscript.
Response: We thank for the reviewer suggestion. The description of drug used and polyesters/linkages are given now at the revised version of the manuscript as Table 2 (highlighted).
4. The challenges and further directions about this field should be in-depth discussed during revisions. It is very important for a review manuscript.
Response: We thank for the reviewer suggestion. The challenges on the field are described along the manuscript at each specific section and the future directions are point out at the “Conclusions section” of the manuscript such as:
“These new stimuli-responsive polyesters were employed as biomaterials in several fields such as in surgery, tissue repair and regeneration, tissue engineering and sustained drug delivery systems for various kinds of bioactive molecules demonstrating their broad applicability, their success and the generally bright future of the field of stimuli-responsive polyester”
“…..progresses in the field are still necessary with more studies and methods for better understanding of polyester degradation characteristics as well for the development of novel degradable polyester devices for actual medical and pharmaceutical application challenges. With the enormous possibility of monomers selection and polymerization processes a multitude of polyesters can be designed with control over degradation, biocompatibility and physical properties to obtain diverse materials for biomedical applications. Furthermore, novel materials with several different responsiveness’s can be created by the combination of different monomers which holds promises for several different applications. As demonstrated just from one small sort of responsive polyesters described herein, they offer a facile route to a plethora of materials with a variety of thermal, degradation and mechanical properties.”
“Altogether, the development and production of synthetic selectively degradable polymers for applications as biomaterials is of upmost importance for the advancement of effective biomedical therapeutics and the future of the field.“
